# Where To Start?
# Transferring Simple Skills to Complex Environments

**Vitalis Vosylius and Edward Johns**
The Robot Learning Lab
Imperial College London
United Kingdom
{vitalis.vosylius19, e.johns}@imperial.ac.uk

**Abstract:**
Robot learning provides a number of ways to teach robots simple skills, such as grasping. However, these skills are usually trained in open, clutter-free environments, and therefore would likely cause undesirable collisions in more complex, cluttered environments. In this work, we introduce an affordance model based on a graph representation of an environment, which is optimised during deployment to find suitable robot configurations to start a skill from, such that the skill can be executed without any collisions. We demonstrate that our method can generalise *a priori* acquired skills to previously unseen cluttered and constrained environments, in simulation and in the real world, for both a grasping and a placing task. Videos are available on our project webpage at https://www.robot-learning.uk/where-to-start.

**Keywords:** Robot manipulation, planning, obstacle avoidance

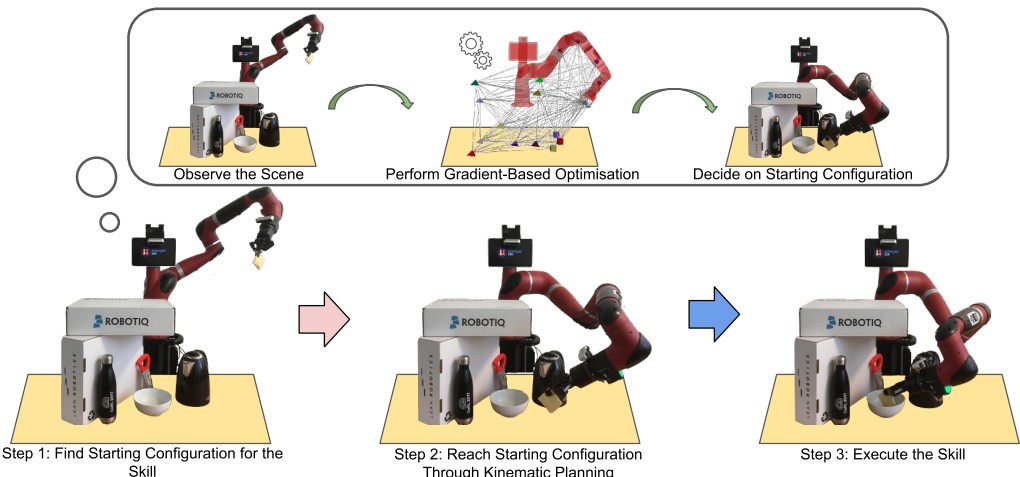

Figure 1: Overview of the proposed framework for one of our considered tasks - placing a held object into a novel container (white bowl). First, the robot observes the scene using head and wrist-mounted cameras. Then, it finds a good configuration to start an *a priori* learnt skill, by optimising the learnt heterogeneous graph-based affordance model. From this configuration, the robot should be able to complete the task without colliding with obstacles. The starting configuration is then reached via kinematic planning, after which the placing skill is executed.

## 1 Introduction

There are many methods for acquiring simple robot manipulation skills, such as grasping, placing or insertion. Such skills can be learnt through methods such as imitation learning [1] or reinforce-

6th Conference on Robot Learning (CoRL 2022), Auckland, New Zealand.

ment learning [2]. However, since these methods are data-driven, they would require training across a huge range of environments if the task is to execute the skills whilst avoiding collisions with the environment. And since this is typically impractical, these skills are typically trained in open, unconstrained environments, without enabling transfer of the skill to more complex, cluttered environments. This then raises the following question: *how do we maintain the high performance of a skill that was trained in obstacle-free environments, whilst ensuring that it does not collide with obstacles in new cluttered environments?*

In complex environments, reactively avoiding obstacles by adding small control adjustments to a learned skill might not be enough, resulting in a need to completely alter the skill and re-learn how to complete the task in the new environment. Therefore, in this paper, we propose to address this by training a separate model to find the optimal robot configuration from which executing the *a priori* acquired skill would complete the task, whilst avoiding collisions with the environment. The model we propose is a novel heterogeneous graph-based affordance model [3], which allows us to represent the target object, the obstacles, and the robot, all in the same Cartesian space, and thus exploit the underlying structure of the learning problem. During deployment, given an initial observation of the environment, we first maximise the predicted affordance score with respect to the robot configuration using gradient-based optimisation, to find the optimal robot configuration from which to start a skill (step 1, Figure 1). The robot can then reach this configuration using motion planning (step 2), and finally execute the skill to complete the task (step 3).

We evaluate our proposed three-step approach on both a grasping and a placing task, in challenging, cluttered and constrained environments, both in simulation and in the real world. We compare it with a single-step end-to-end approach, where skills are trained in cluttered environments directly, and we also study alternative ways to find the optimal starting configuration. We also carry out an ablation study to evaluate the benefits of using a heterogeneous graph as the input of our affordance model. Overall, our experiments validate our method as an effective way to transfer simple *a priori* skills to complex environments with previously unseen clutter.

## 2 Related Work

**Robotic Manipulation in Cluttered Environments**. Methods that address robotic manipulation in environments where collisions with the scene are undesired have primarily focused on the grasping task [4, 5, 6, 7, 8]. Most works try to predict a stable grasping pose for which the end-effector would not collide with the surrounding objects without taking the complete configuration of the robot into account. [9] learns latent plans and when to switch to a pre-trained target-driven policy. These plans can be sampled, scored and decoded into end-effector policy actions that avoid objects on the tabletop and grasps the target. On the other hand, we consider the complete configuration of the robot and reach an optimal starting configuration using kinematic planning.

**Optimising Learnt Models at Test Time**. Recently, many works have focused on optimising learnt models with respect to their inputs at inference to perform various robotic manipulation tasks [10, 11]. [12] learns an energy function from demonstrations that, when minimised at inference, can produce actions for a closed-loop policy. In [13], Cartesian robot representation and joint angles are used as an input, letting the model decide the most useful representation. [14, 15] tackles the grasp synthesis problem by optimising the learnt success probability model at inference. [16] has used learnt models as constraints on the key configuration in a nonlinear trajectory optimisation framework. We take a similar approach but use a learnt affordance model to find a robot configuration from which a specific skill is likely to succeed.

**Representation Learning for Robot Manipulation**. A large amount of work has focused on finding helpful robot and scene representations. Semantic keypoints representing objects proved to be beneficial in various manipulation tasks [17, 18, 19]. [20, 21] learns dense object descriptors that map raw input images to pixel-wise object representations. Recently, representing the scene using graphs and utilising graph neural networks has been extensively explored [22, 23, 24]. On the other hand, complex systems with various actors have been successfully modelled using heterogeneous graphs exploiting their underlying structure [25, 26]. Examples include academic graphs [27, 28], recommendation systems [29, 30], and LinkedIn economic graphs [31]. In this work, we also try to exploit the problem's underlying structure using heterogeneous graphs.

# 3 Method

## 3.1 Overview

We address robot manipulation in cluttered and constrained environments using the three-step approach shown in Figure 2 (left). We assume access to a skill (e.g. grasping, placing, pushing) which has already been acquired (e.g. through imitation learning or reinforcement learning), and during deployment, we assume a specification of which object to execute the skill on (e.g. through a high-level planner). Throughout this paper, we call this object *the target*. In our experiments we test our method on two skills, grasping and placing, which we train in simulation using behavioural cloning. However, our method can work with any *a priori* acquired skill and is agnostic to how that skill was acquired. Therefore, we now use the general notation $\pi(.)$ to represent a function that executes a given skill.

Our work aims to position a robot in such a way that, even in cluttered and constrained environments, these *a priori* acquired skills can be successfully executed without colliding with the surrounding environment. We do so by learning an affordance model, for each individual skill. Given an initial observation of the scene and a robot configuration, this model predicts whether the skill would be successful if started from this configuration. By maximising the predicted affordance score with respect to the robot's joint angles during deployment, a starting configuration for a particular skill can be found and reached using motion planning.

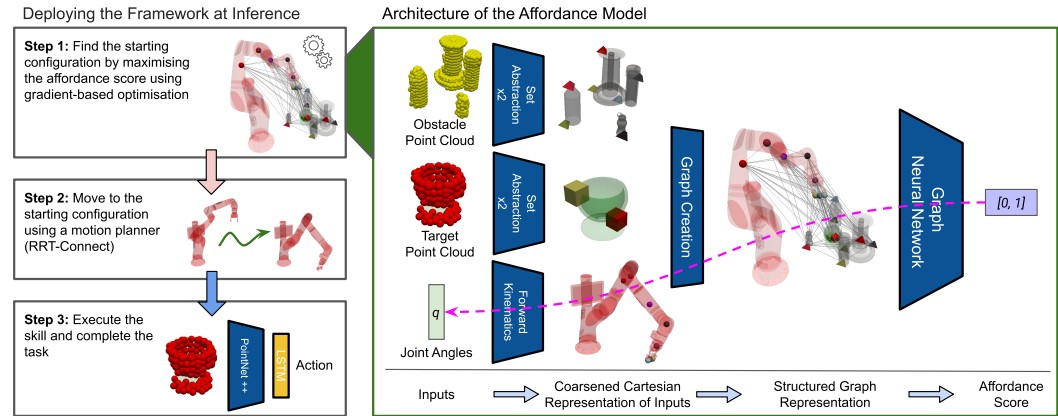

Figure 2: (Left) Overall control-flow of our framework during deployment. (Right) Architecture used to learn the affordance model using segmented point clouds and robot joint angles ($q$). Here, the magenta dashed line represents the gradient flow used to optimise the robot joint angles by maximising the affordance score.

## 3.2 Affordance Model

We assume that even in a cluttered environment, there exists a distribution of robot configurations from which an *a priori* learnt skill would be successfully executed, i.e. $Y = 1$. To find such configurations, we introduce an affordance model $p_\theta(Y = 1 | \mathcal{P}_{target}, \mathcal{P}_{obs}, q)$ for each skill $\pi(.)$. Given an initial observation of the target $\mathcal{P}_{target}$ and its surrounding obstacles $\mathcal{P}_{obs}$, and the robot configuration $q$, the model predicts the probability of the robot successfully executing the skill from this configuration without colliding with any obstacle. We use segmented point clouds to represent $\mathcal{P}_{target}$ and $\mathcal{P}_{obs}$, and assume we know which policy to execute and which object in the scene is the target. Note that considering only the pose of the end-effector in most cases is not sufficient for this problem, and we need to consider the complete robot configuration (i.e. joint angles).

### 3.2.1 Structured Graph Representation

To accurately learn $p_\theta(.)$, the model needs to reason about the target, obstacles, robot's configuration, and the relative position of each of these. It also needs to capture the capabilities of skills and any low-level controllers used to execute them. While *a priori* acquired skills can be designed to work

on different robots, the affordance model is dependent on the specific robot and its kinematics as it needs to reason about the robot's geometry and its movement during skill execution. Instead of compressing the scene observation into a single vector and relying on a neural network to interpret the vector correctly, we want to exploit the underlying structure of this problem. We do so by jointly representing the target, obstacles and the robot as a heterogeneous graph.

Heterogeneous graphs allow to attach relevant information to different types of nodes and define structured relations between them. This allows to systematically capture numerous semantic relationships across multiple types of objects. More formally, a heterogeneous graph can be represented as a tuple $\mathcal{G} = (\mathcal{V}, \mathcal{E}, \mathcal{F})$, where $\mathcal{V}$ is a node set, $\mathcal{E}$ is an edge set, and $\mathcal{F}$ is a node feature set. It also has the associated node type mapping function $\phi : \mathcal{V} \rightarrow \mathcal{A}$ and the edge type mapping function $\psi : \mathcal{E} \rightarrow \mathcal{R}$, $|\mathcal{A}| + |\mathcal{R}| > 2$. Here $\mathcal{A}$ and $\mathcal{R}$ are node and edge types, respectively. For more detailed explanation of heterogeneous graphs and graph neural networks we refer the reader to [32, 33, 34].

While the target and the obstacles in the environment can be represented in Cartesian space as point clouds, the robot configuration $q$ inherently lies in a different (joint angle) space. The highly non-linear mapping between the two spaces can make capturing the desired distribution extremely difficult. We, therefore, represent the complete robot configuration not as a list of joint angles but rather as a set of positions of predefined key points on the robot that can be calculated using fully differentiable Forward Kinematics. These robot key points can, for example, be the centre of mass for each link of the robot. To differentiate between robot key points in an unordered set, each has a feature vector containing a one-hot encoding to identify it.

Using the entire point clouds in the heterogeneous graph would result in a massive structure hindering the learning process. We, therefore, use Pointnet++ set abstraction layers [35] to coarsen the point cloud by sampling a fraction of points and locally encoding their neighbourhood. Together with the robot's key points, we now have three sets of points together with their local feature vectors $(\{pos_t, \mathcal{F}_t\}, \{pos_o, \mathcal{F}_o\}, \{pos_r, \mathcal{F}_r\})$, representing the target, obstacles and the robot, respectively. We use these to create a fully connected heterogeneous graph that jointly represents the robot, the target object and the obstacles in the environment (see Figure 2 right). We use edge features to represent the relative transformations between node pairs. This way, the network never sees any absolute position, which allows for an easy generalisation through the task space. These relative transformations are differentiable, which still allows us to take gradients with respect to the absolute positions of the robot key points (Figure 3 c). The gradient of each key point shows in which direction it should be moved to increase the affordance prediction. Using kinematic Jacobian and aggregating gradients from the key points, we can compute the gradient with respect to the robot's joint angles (magenta dashed line in Figure 2) and use it in a gradient based-optimisation framework.

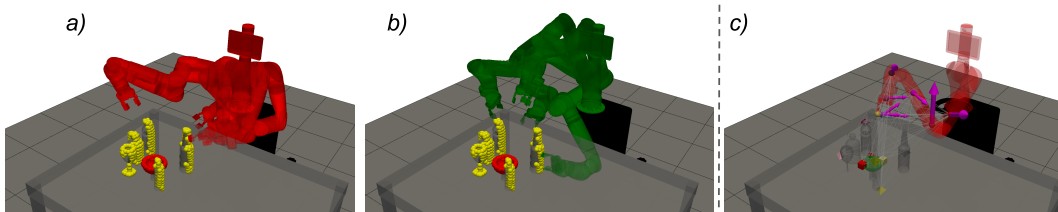

Figure 3: Examples of robot configurations that, when a grasping skill is executed, will lead to (a) a collision with the environment or (b) successful task completion. Yellow and red point clouds represent the obstacles and the target, respectively. (c) A heterogeneous graph representation used and gradients with respect to the positions of the key points defined on the robot (magenta arrows).

### 3.2.2 Training

We use a graph neural network to learn $p_\theta(.)$ from the structured input presented in Section 3.2.1. The whole neural network architecture we use to learn $p_\theta(.)$ can be seen in Figure 2 (right) and is trained end-to-end. More details on the architecture are presented in the supplementary material.

To train this network, we use a dataset composed of positive and negative samples created by rolling out the learnt skills from the different configurations and recording the outcome. We do these rollouts in different types of procedurally generated simulated scenes with varying amounts of clutter (see Figure 4). One roll-out creates one sample, consisting of the starting robot configuration $q_i$,

initial scene observation $O_i$, and the outcome $y_i$: $D_{rollouts} = \{q_i, O_i, y_i, \}_{i=1}^N$. We choose where to start the *a priori* acquired skill at random such that the wrist camera is pointing toward the object, and the robot is not colliding with the environment. We validate the task's success using privileged information available in simulation and assign a label $y$ to the roll-out accordingly (0 or 1). Any roll-out leading to a collision is treated as a failure and assigned the label 0. In our implementation, the initial scene observation $O_i$ is composed of segmented point clouds ($\mathcal{P}_{target}$ and $\mathcal{P}_{obs}$) aggregated from two-depth cameras mounted on the robot's head and wrist (see Figure 1). Using multiple cameras allows for better scene coverage, which can be crucial when estimating affordance scores from initial scene observation. Figure 3 (*a* and *b*) shows examples of robot configurations in a cluttered environment used as negative and positive samples, respectively. We add noise to all the depth images used to obtain point clouds according to [36] and distort the ground truth segmentation maps from simulation to better mimic a real-world setting.

Having both negative and positive examples, we use a Binary Cross-Entropy loss function to train the network as an affordance classifier. Additional details regarding the affordance model and its training are presented in the supplementary material.

### 3.3 Deployment

The first step of our method during deployment is to find an optimal robot configuration from which to start an *a priori* acquired skill using initial scene observation. We do so by optimising the learnt affordance model with respect to the robot's configuration to maximise the probability of a particular skill being successful. Optimising the learnt model at inference allows us to include any other task or safety-relevant analytical constraint. We utilise this and add joint limit and additional collision avoidance ($g_{coll}(.)$) constraints between each link ($l$) in a set of links comprising the robot ($L$) and known obstacles ($obs_j$) in the environment ($E$). This results in a constrained, non-convex optimisation problem (Equation 1) which we solve using a gradient-based IPOPT optimiser [37].

$$\begin{aligned} \max_q \quad & p_\theta(Y = 1 | \mathcal{P}_{target}, \mathcal{P}_{obs}, \pi_k, q) \\ \text{s.t.} \quad & q_{min} \geq q \geq q_{min} \\ & g_{coll}(FK_l(q), obs_j) \geq 0, \forall l \in L; \forall j \in E \end{aligned} \tag{1}$$

In our implementation, known obstacles ($E$) include the table and bounding boxes of segmented objects as cuboid obstacles. This is a coarse estimation of the environment later used by a motion planner. Thus, including $g_{coll}(.)$ ensures that the solution is not considered to be in a collision by the motion planner, which would make finding a collision-free path to this configuration infeasible. In our experiments, solving Equation 1 took on average 3.2 seconds and had to be done only once using the initial scene observation. Once the joint angles from which the low-level skill should start are found, we use an RRT-Connect [38] motion planner to find a collision-free path to reach that configuration (second step of our framework). Finally, the third step is to execute the *a priori* acquired skill completing the task.

## 4 Experiments

### 4.1 Experimental Setup

We evaluated our method on two manipulation tasks in scenes with varying amounts of clutter: 1) grasping novel objects and 2) placing a held object inside novel containers. For the grasping task, the robot aims to grasp the target object stably. For the placing task, the robot starts with a small object inside its gripper, and the goal is to place it inside a target container (e.g. a bowl). In both cases, collision with obstacles is considered a failure, making the tasks extremely challenging.

We acquire the skill to solve each of these two tasks respectively using Behaviour Cloning (BC) in simulation only. We use various everyday objects from ShapeNet [39] and expert trajectories created by an oracle planner. The skills output 6D end-effector velocities and use observations from the wrist-mounted depth camera in terms of the segmented point cloud of the target object. Our BC policy networks consist of PointNet++ [35] followed by an LSTM cell (see Figure 2, bottom left). Our closed-loop skills are trained in a tabletop environment with no obstacles. The supplementary material provides a more detailed description of our BC skills and how we train them.

In simulation, we procedurally generate six types of randomised environments with inherently different structures containing the target object and surrounding obstacles (see Figure 4), using the CoppeliaSim [40] simulator with PyRep [41]. *Env 1* is the easiest with no obstacles and serves as a baseline showcasing the abilities of learnt skills in isolation. The following environments keep increasing in complexity. *Env 2* includes obstacles (primitive shapes and objects from ShapeNet) surrounding the target object. *Env 3* and *5* have structured obstacles representing drawers and shelves, while environments *4* and *6* add to complexity of *3* and *5* respectively with additional obstacles.

We create $D_{rollouts}$ for each type of environment individually by rolling out skills from 10 different configurations per randomised scene, totalling 500 samples per type of environment. For a fair comparison, all evaluated methods use the same observation space (segmented point clouds) processed by PointNet++. Moreover, we use the same number of training examples to train all the methods and roughly match the number of trainable parameters. Baselines trained end-to-end in cluttered scenes use additional data matching the number of training samples used to acquire our BC skills.

In all environments, we randomise obstacles, their shapes and the overall arrangement. In simulation, we evaluate all method on 500 randomised environments of each type, where we randomly choose a target object from 100 objects from ShapeNet [39] that have not been seen during training. All evaluation environments are also validated using an oracle planner to ensure there is at least one collision-free way to complete the task. We run all experiments with different random seeds six times and present the mean and standard deviation of the success rate. In the supplementary material, we provide results in the graph form showing the trends more clearly.

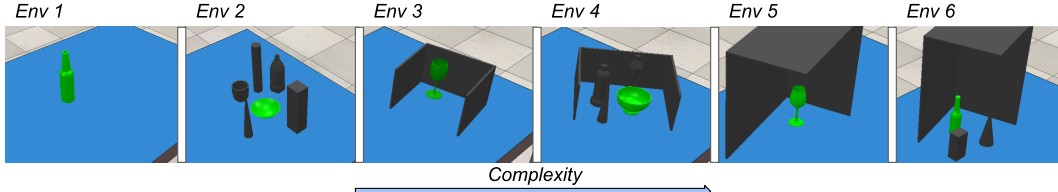

Figure 4: Examples of different types of procedurally generated randomised simulated environments. Inherently different structures introduce a different complexity in each type of environment.

## 4.2 Comparison Against Learning a Single End-to-End Policy

Our first set of experiments aims to answer the following question: in what type of environments does our three-step approach lead to the most noticeable performance gains over learning a single end-to-end policy directly in cluttered environments?

To answer this, we compare the performance of our method with two fully end-to-end Behaviour Cloning baselines: *BC(eef)* and *BC(joint)*. Contrary to the skills described in Section 4.1, these policies are trained using demonstrations from scenes with obstacles and need to learn both to avoid obstacles and to complete the task. Generally, this requires controlling the full configuration of the robot. We therefore concatenate current robot joint angles to every point in the point cloud. *BC(eef)* operates in a 6D end-effector action space, whilst *BC(joint)* uses a joint angle action space. In this set of experiments, all methods are trained using data from each type of environment separately, and evaluated on unseen configurations of these environment types. The results are shown in Table 1.

| Method | Env 1 | | Env 2 | | Env 3 | | Env 4 | | Env 5 | | Env 6 | |
| --- | --- | --- | --- | --- | --- | --- | --- | --- | --- | --- | --- | --- |
| | Grasp | Place | Grasp | Place | Grasp | Place | Grasp | Place | Grasp | Place | Grasp | Place |
| BC (eef) | 79.1 ± 3.7 | 89.5 ± 2.3 | 43.7 ± 3.6 | 56.4 ± 4.8 | 41.5 ± 3.6 | 47.4 ± 4.0 | 38.0 ± 2.9 | 41.6 ± 4.6 | 11.0 ± 4.4 | 12.1 ± 1.7 | 5.5 ± 2.1 | 9.8 ± 2.9 |
| BC (joint) | 15.6 ± 5.8 | 21.1 ± 7.8 | 11.3 ± 7.0 | 17.9 ± 7.3 | 5.4 ± 3.0 | 7.0 ± 3.0 | 0.0 ± 0.0 | 2.7 ± 0.9 | 0.0 ± 0.0 | 0.0 ± 0.0 | 0.0 ± 0.0 | 0.0 ± 0.0 |
| Ours | **82.6 ± 2.1** | **91.0 ± 2.0** | **79.8 ± 1.4** | **89.1 ± 1.7** | **82.4 ± 2.2** | **90.0 ± 1.2** | **79.1 ± 1.9** | **84.7 ± 1.5** | **80.2 ± 3.8** | **83.0 ± 3.2** | **75.1 ± 1.8** | **78.6 ± 1.4** |

Table 1: Success rates (mean and standard deviation) of our 3-step method compared to single-step policies trained end-to-end in constrained environments directly.

*BC(joint)*, which uses a joint-angle action space, does not yield good results even in environments without obstacles. This is expected due to highly non-linear mapping between the joint angle space and Cartesian space, in which point clouds are represented. *BC(eef)* performs better but struggles in more cluttered environments. Our method, on the other hand, performs well in all environments, showing only a minor performance decrease with increasing cluttering. Therefore, to answer our

initial question, the more complex the environment the more noticeable the performance gains of our method over learning a single end-to-end policy.

## 4.3 Different Ways of Finding Starting Configurations

In this second set of experiments, we aim to answer the following question: is optimising the learnt affordance model a better way to find the policy's starting configuration than alternatives?

The alternatives (or baselines) we test include 1) a *Naive* approach, where a random configuration with the wrist camera pointing at the target object is found and reached using a motion planner, 2) a *Generative* approach, which generates a set of joint angles using a trained CVAE [42], conditioned on a segmented point cloud, 3-4) two-stage Behaviour Cloning (*BC2(eef)* and *BC2(joint)*) approaches using different action spaces, where a closed-loop policy is learnt from demonstrations to reach the closest validated starting configuration and execute the skill. The naive baseline serves to validate the need to learn these starting configurations. The generative baseline allows us to validate our way of obtaining them (regressing joint angles vs optimising affordance score). Finally, the *BC2* baselines aim to validate the added benefit of the second step in our framework - reaching a starting configuration using a motion planner. To better showcase the generalisation across unseen clutter, we train all methods using data from 5 of the 6 environment types and evaluate them on the remaining type. The results can be seen in Table 2.

| Method | Env 1 | | Env 2 | | Env 3 | | Env 4 | | Env 5 | | Env 6 | |
|---|---|---|---|---|---|---|---|---|---|---|---|---|
| | Grasp | Place | Grasp | Place | Grasp | Place | Grasp | Place | Grasp | Place | Grasp | Place |
| Naive | 81.1 ± 1.9 | **93.0 ± 1.9** | 51.8 ± 5.7 | 68.8 ± 2.6 | 46.2 ± 5.8 | 60.3 ± 4.8 | 33.5 ± 7.3 | 50.2 ± 2.2 | 32.3 ± 4.8 | 43.1 ± 3.2 | 29.5 ± 8.6 | 30.0 ± 4.3 |
| Generative | 52.1 ± 20.6 | 65.2 ± 11.1 | 59.7 ± 9.5 | 62.4 ± 7.7 | 42.8 ± 11.0 | 46.8 ± 13.7 | 23.7 ± 6.8 | 34.0 ± 8.9 | 12.8 ± 5.9 | 19.9 ± 6.1 | 4.2 ± 3.5 | 11.2 ± 4.6 |
| BC2 (eef) | 81.0 ± 1.5 | 88.5 ± 2.6 | 63.5 ± 3.9 | 70.4 ± 2.9 | 63.1 ± 5.1 | 67.5 ± 9.2 | 58.2 ± 5.7 | 65.2 ± 8.2 | 39.9 ± 3.6 | 44.4 ± 4.1 | 33.3 ± 3.6 | 25.4 ± 3.5 |
| BC2 (joint) | 42.7 ± 11.5 | 56.2 ± 6.2 | 40.5 ± 8.4 | 43.2 ± 7.2 | 37.0 ± 7.0 | 39.3 ± 5.0 | 25.6 ± 8.0 | 32.9 ± 10.5 | 8.2 ± 2.3 | 11.7 ± 1.9 | 3.2 ± 1.9 | 8.0 ± 3.1 |
| Ours | **81.8 ± 2.1** | 91.2 ± 2.1 | **81.0 ± 1.9** | **88.0 ± 1.8** | **79.7 ± 1.7** | **89.4 ± 1.9** | **79.9 ± 3.6** | **81.7 ± 3.9** | **77.1 ± 4.2** | **79.0 ± 3.2** | **74.5 ± 4.0** | **76.5 ± 5.0** |

Table 2: Success rates (mean and standard deviation) of our 3-step method compared to other relevant multi-step approaches.

As expected, the *Naive* approach performs well in an uncluttered environment, but unlike our method, its performance significantly decreases with increasing clutter. Other baselines also struggle with cluttered environments, especially the ones operating directly in the joint angle space. They often start the skill execution when the wrist camera cannot see the target. Together, these results provide evidence of the added value of our method compared to the baselines.

## 4.4 Ablations

Our third set of experiments investigates how important the structured input representation we use is for the affordance model's performance. To do this, we train two affordance models that take as input a simple concatenation of the segmented point cloud representing the environment (obtained using PointNet++) and the robot's configuration, in terms of joint angles (*PointNet(joint)*) and Cartesian key points (*PointNet(Cartesian)*) respectively. We then use this joint representation to predict the affordance score. Additionally, to validate our claim that it is crucial to consider the full configuration of the robot, we train a version of our proposed network that uses only key points on the end-effector to represent the robot's configuration (*Ours (eef only)*). We evaluate these methods the same way as in Section 4.3. Table 3 shows the results of this set of experiments.

*PointNet(joint)* and *PointNet(Cartesian)* did not perform well, showing that a structure input representation is really important. Using only key points representing the robot's end-effector instead of the whole robot, on the other hand, yielded better results in *Env 1*. However, this approach did not perform well in more cluttered and constrained environments, where it is crucial to consider the whole configuration of the robot to avoid collisions and complete the task.

| Method | Env 1 | | Env 2 | | Env 3 | | Env 4 | | Env 5 | | Env 6 | |
|---|---|---|---|---|---|---|---|---|---|---|---|---|
| | Grasp | Place | Grasp | Place | Grasp | Place | Grasp | Place | Grasp | Place | Grasp | Place |
| PointNet(joint) | 45.2 ± 3.5 | 51.5 ± 4.6 | 40.9 ± 7.8 | 46.5 ± 7.9 | 36.7 ± 6.5 | 38.1 ± 5.5 | 24.9 ± 5.6 | 29.9 ± 7.3 | 6.3 ± 2.7 | 12.3 ± 3.8 | 4.4 ± 1.4 | 15.0 ± 3.4 |
| PointNet(Cartesian) | 39.4 ± 5.6 | 46.0 ± 7.9 | 34.0 ± 6.6 | 36.0 ± 8.7 | 32.4 ± 5.1 | 35.4 ± 10.5 | 20.2 ± 5.3 | 24.2 ± 7.2 | 4.7 ± 1.6 | 17.2 ± 3.6 | 5.0 ± 1.2 | 9.6 ± 2.7 |
| Ours (eef only) | **82.7 ± 1.6** | **92.1 ± 1.2** | **75.1 ± 2.4** | **81.8 ± 4.2** | **61.5 ± 4.4** | **67.2 ± 9.4** | **62.2 ± 6.4** | **64.1 ± 4.9** | **39.5 ± 7.6** | **54.9 ± 7.5** | **40.3 ± 5.9** | **53.3 ± 7.4** |

Table 3: Success rates (mean and standard deviation) of our 3-step method using different representations of the scene and the robot to learn the affordance model.

## 4.5 Real World Evaluation

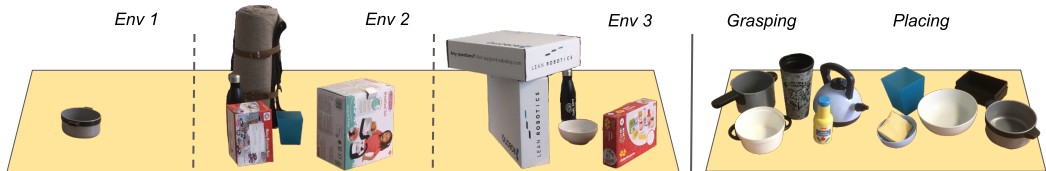

Figure 5: (Left) Examples of different types of environments used in real-world experiments. (Right) Objects used to evaluate grasping and placing tasks.

Although our method is trained entirely in simulated environments, it uses noisy observations and distorted segmentations and should be able to transfer to the real world. To validate this, we conduct real-world experiments involving a Sawyer robot equipped with a Robotiq F2-85 gripper and head and wrist-mounted Intel Realsense D435 depth cameras (see Figure 1). We

|        | Env 1 | | Env 2 | | Env 3 | |
|--------|-------|-------|-------|-------|-------|-------|
| Method | Grasp | Place | Grasp | Place | Grasp | Place |
| Naive  | 9 / 15 | 12 / 15 | 6 / 15 | 7 / 15 | 2 / 15 | 5 / 15 |
| Ours   | 11 / 15 | 12 / 15 | 10 / 15 | 11 / 15 | 10 / 15 | 10 / 15 |

Table 4: Success rates of our method and the *Naive* baseline in a real world setting.

evaluate our method in three types of environments with different levels of clutter and five objects unseen during training for both grasping and placing tasks (see Figure 5). We run three evaluations per object for each type of environment, randomising the scene's configuration. For segmentation, we use a method for unknown object segmentation [43] and manually specify which object is to be manipulated. The results of the real-world experiments can be seen in Table 4. Our method, trained solely in simulated environments, was able to directly adapt to all types of challenging real-world environments without any major decrease in performance. In comparison, and consistently with the results of our second experiment, the *Naive* baseline struggled to deal with more complex environments (see Section 4.3).

## 5 Limitations, Failure Modes and Future Work

Although our method shows promising results in generalising learned skills to unseen cluttered environments, it has its limitations. First, for skills trained in reality rather than simulation, collecting data for the affordance model could be cumbersome. Second, our learnt affordance model would need to be re-trained every time a skill changes. Third, relying on perception pipelines for segmentation can lead to failures. Fourth, optimisation of the learnt affordance model is quite sensitive to the initialisation provided and can get stuck in bad local minima. And finally, our method may fail in cases where executing the skill without contact with the environment is impossible, such as with long-horizon skills that require significant movement of the whole robot. However, the majority of manipulation skills acquired using methods such as Imitation Learning or Reinforcement learning are simple and short-horizon. In future work, we plan to explore and address these limitations, study the applicability to other tasks beyond grasping and placing, and extend our approach to longer-horizon tasks by chaining together multiple skills, aiming towards a full pick-and-place system.

## 6 Conclusions

In this paper, we proposed a method for the robotic manipulation of objects in cluttered and constrained scenes, where collision with the environment is undesired. We use an approach that combines *a priori* acquired skills with the use of an affordance model of our own design. Optimising this affordance model during deployment allows us to find suitable configurations from which starting a given robotic manipulation skill would lead to its successful completion without collision with the environment. To efficiently learn this affordance model, we introduced a novel, heterogenous graph-based representation that jointly captures information about the target object, scene obstacles, and the robot itself, in a structured way. We showed that our method outperforms various baselines, can generalise to unseen cluttered and constrained scenes, and can transfer from simulation to reality.

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
