# OpenReview forum: "Where To Start? Transferring Simple Skills to Complex Environments"
_robot-learning.org/CoRL/2022/Conference — CoRL 2022 Poster_

### Official Review · Reviewer_oZZf · 2022-07-23

**Originality:** Fair
**Technical Quality:** Fair
**Clarity Of Presentation:** Very Good
**Impact:** 3

**Recommendation:**

Weak Reject: I recommend rejecting the paper, but will not argue for my recommendation if the majority of other reviewers have a different opinion.

**Summary:**

The authors proposed a graph neural network model to determine the initial robot configuration in a cluttered environment. This initial configuration is then deployed into a motion planning model to solve the collision-free manipulation task. The authors show in both simulations and in the real world how this model can help improve the success rate of skill transfer.

**Issues:**

1. The proposed method tries to learn an initial configuration to simplify the subsequent manipulation task. However, reaching this configuration could already be problematic, especially when the environment has a very narrow feasible space. When a simple motion planner is used to reach the initial configuration as described by the authors, why not use a motion planner to solve the whole problem?
2. How the downsampling of the point cloud could encode the geometry of the environment? What is the local feature vector? Why it’s necessary to learn the affordance model? It seems to me that the model only needs a subset of the point cloud to compute the transformation.
3. What is $y_i$ in section 3.2.2?
4. In the deployment, the author also added a collision avoidance constraint. Why is this constraint needed? Shouldn’t the model learn it?
5. I think the joint limit constraint could be parameterized differently, such as a sigmoid function [1]. In such a way, you could avoid the constraints in the optimization.
6. It would be more beneficial to compare with some proper baselines such as [1], [5].

[1] Ratliff, N.D., et al, Riemannian motion policies. arXiv preprint arXiv:1801.02854 (2018)

[2] Finean, et al. "Predicted composite signed-distance fields for real-time motion planning in dynamic environments." *Proceedings of the International Conference on Automated Planning and Scheduling*. Vol. 31. 2021.

[3] Liu, Puze, et al. "Regularized Deep Signed Distance Fields for Reactive Motion Generation." *arXiv preprint arXiv:2203.04739* (2022).

[4] Todorov, Emanuel. "Compositionality of optimal control laws." *Advances in neural information processing systems* 22 (2009).

[5] Johannink, Tobias, et al. "Residual reinforcement learning for robot control." *2019 International Conference on Robotics and Automation (ICRA)*. IEEE, 2019.

**Quality Of The Limitations Section:**

Additional details required

**Reviewer Expertise:**

5: The reviewer is absolutely certain that the evaluation is correct and very familiar with the relevant literature

**Robotics Focus:**

Sufficient demonstration on hardware

**Strengths And Weaknesses:**

The document is generally well written and easy to understand.

The numbers clearly illustrate the proposed methodology.

The number of random seeds used for statistical analysis is insufficient.

I have some doubts about the scalability and generalizability of the proposed approach. The proposed model is trained in a specific scenario with limited deviations. The proposed approach has also been validated in a similar setting. In addition, the affordance model is also conditioned on pre-trained skills. All these settings limit applicability to a broader range of tasks and environments.

This paper lacks rational motivation. The paper's objective is to address the problem of collision avoidance for manipulation tasks. Many different methods already exist to deal with unstructured environments, such as the primitive approximations [1] or the SDF-based approaches [2, 3], so why is the proposed method beneficial compared to these approaches? Also, different approaches exist for skill transfer, such as policy composition [4], and learning residual policies [5]. What is the advantage of only finding the initial configuration?

**Summary Of Recommendation:**

As stated in the strength and weaknesses, I think the paper does not have a clear motivation.
The scalability and generalizability are not clear to me. I would appreciate it if the author could conduct more experiments on it.
The advantage and drawbacks are not well discussed. The experiment runs only 3 seeds which are not statistically sufficient.
I would suggest rejecting the paper in its current status.

---

> ### Author Response · Authors · 2022-08-26
> **Response to the Reviewer oZZf**
>
> We would like to thank the reviewer for taking the time to review our submission and for all the helpful suggestions for improving the manuscript.
>
> **Statistical significance**
>
> We have increased the number of replications of our simulation experiments using different random seeds to 6, doubling the number in the original manuscript to allow for statistical significance. The trends that were observed before still hold and further validate our claims made in the original manuscript.
>
> **Scalability and generalizability**
>
> We have evaluated our method on a **large number of environments** with configurations **not seen during training**. In all evaluation environments, **the number of obstacles, and their shapes and sizes were randomised**. In addition, in our second set of experiments, we tested our methods in completely different types of environments, with **structures significantly different from the ones used for training**. We did so by holding out one type of environment during training and running evaluation solely in that type of environment. Our method was able to generalise across them using only the initial observation of the environment. We added additional examples of evaluation environments in the revised manuscript (supplementary material) (lines 93-94). Additionally, using skills acquired in unconstrained environments allowed us to retain their generalisability to novel objects without the need to re-learn that in constrained environments. Therefore, we argue that our method is highly scalable since we transfer skills to new environments without having to re-learn those skills.
>
> **Conditioning affordance model on pre-trained skills**
>
> The affordance model needs to know which skill is being executed, as it needs to reason about the robot's trajectory during the skill. However, please note that both of the skills we study (grasping and placing) do generalise across different objects (e.g. our grasping skill is trained to grasp any object). An interesting extension to our method would be to condition the affordance model on the skill identity, e.g. using one-hot encoding.
>
> **Motivation**
>
> Yes, our method addresses the problem of collision avoidance: but it does so for skills which have already been trained in collision-free environments. We are not proposing a method to actually learn those skills, because there are many methods available for that. Instead, our method allows for the transfer of existing skills to new, constrained environments,** without the need to re-train them in these environments or alter them in any way**. The papers which the reviewer mentions are all methods for collision avoidance; they are not methods to take an existing policy and enable it to work in different environments to where it was trained (which is what our method does). For example, another method for collision avoidance would just allow a robot to reach a pose in space whilst avoiding collisions: it would not then allow the robot to grasp an object, or place an object down. We have updated the introduction in the revised manuscript to emphasise our motivation more clearly (lines 21-23).
>
> **Skill transferring methods**
>
> Methods that transfer skills to new environments using, for example, residual policies, can adapt to small changes in the scene by interacting with the new environment. However, in our considered settings where there is significant clutter, **small adjustments to the original policy would not be enough, and the robot configuration would need to change drastically to avoid collisions.** In such cases, a vast amount of interaction data would be required to adjust to the new environments because **the entire policy would need to be re-learned across all these environments**. Our method, on the other hand, does not require to re-learn the policy, but rather positions the robot in such a way that an existing policy can be successful.
>
> **Reaching starting configuration**
>
> We agree that in environments with very narrow feasible spaces reaching the predicted starting configuration can be a challenging problem on its own. However, this is true for all approaches that consider collision avoidance and is an open research question on its own.
>
> **Solving the task using a motion planner**
>
> In general, our method considers manipulation skills that involve **more than just collision avoidance and cannot be solved using just a motion planner. **
> Many complex robotic skills cannot be designed using just a motion planner or other model-based methods, since we do not have access to an accurate model of the environment. Instead, skills are usually acquired using techniques such as Imitation Learning or Reinforcement Learning. In our approach, we use closed-loop Behaviour Cloning policies to acquire such skills, and a motion planner is then used to reach the predicted starting configuration - a part of the task that can be modelled.

---

> > ### Author Response · Authors · 2022-08-26
> > **Response to the Reviewer oZZf (continued)**
> >
> > **Point cloud processing**
> >
> > The point clouds are **not downsampled, but instead are coarsened using Pointnet++ set abstraction layers** [1]. Set Abstraction (SA) layers are composed of three layers: a sampling layer, a grouping layer and a Pointnet layer. In short, SA layers return subsampled pointcloud together with feature vectors summarising their local context. **Local feature vectors are learnt together with the affordance model end-to-end. **
> >
> > The model still uses the whole point cloud as its input. However, the use of SA layers reduces the dimensionality of the observation space before adding the robot's configuration and predicting the affordance score using a GNN. **This allows for more efficient GNN training and results in better generalisation.**
> >
> > [1] C. R. Qi, L. Yi, H. Su, and L. J. Guibas. Pointnet++: Deep hierarchical feature learning on point sets in a metric space. Advances in neural information processing systems, 30, 2017
> >
> > **Used notation**
> >
> > yi is a label assigned to the roll-outi, where yi=0 and yi=1 mean roll-out was successful or not, respectively. We have updated the manuscript to more clearly define the notation used in section 3.2.2 (lines 143-144).
> >
> > **Collision avoidance constraint**
> >
> > The model should learn to produce a collision-free configuration without the need for additional collision avoidance constraints. However, a motion planner needs to find a collision-free path to the produced configuration. We do not have access to the ground truth model of the environment that motion planners could use, but rather we use a very coarse estimation of the obstacles in the environment in terms of bounding boxes of segmented objects as cuboid obstacles obtained from the initial observation. Due to this, a collision-free robot configuration found using the affordance model can be still treated as being in a collision by the motion planner. This would make it infeasible for the motion planner to find a collision-free path from starting configuration to goal one as the latter is already in a collision. Adding collision constraints between the robot and this coarse estimation of the environment mitigates the discussed issue without requiring the ground truth model.
> >
> > **Parametrisation of joint limit constraint**
> >
> > We agree that it is a nice and elegant way to parameterise joint limits to avoid constraints in the optimisation, and thank you for the suggestion. However, because we are already solving the constrained optimisation problem and doing it only once using the initial scene observation, it does not create an excessive amount of computational overhead.
> >
> > **Additional baselines**
> >
> > The proposed baselines could allow for the skills to adapt to small changes in the scene and in scenarios where small adjustments to the original policy are not enough would need to re-learn the skill itself. This is not the focus of our work, as we are trying to directly transfer old skills to new environments without the need to re-learn or alter them. We have extended the discussion and justification regarding the chosen baselines in the revised manuscript (supplementary material)(lines 102-104 and 122-125).

---

### Official Review · Reviewer_jW3a · 2022-07-30

**Originality:** Very Good
**Technical Quality:** Very Good
**Clarity Of Presentation:** Excellent
**Impact:** 4

**Recommendation:**

Strong Accept: I recommend accepting the paper and will argue for my recommendation even if other reviewers hold a different opinion.

**Summary:**

The paper presents a approach that can provide suitable starting poses for a priori known skills.
Skills in the paper are for grasping and placing tasks, but there is not reason apparent why it should
not generalize to other skills. The approach can facilitate robot manipulation in cluttered environments
The learned model can be optimized at runtime to adapt to current environments. The authors propose the
heterogeneous graph-based representation.
Evaluation is done in simulation and on a real robot to both show the need of the component and its performance
compared to alternative approaches.

**Issues:**

- There is no mentioning of any time requirements for training, learning, or optimization at test time. This information
would be of great interest and a crucial detail to judge the performance and usability of the approach.

- A bit extended discussion and details in the limitations section would improve the paper.

- I might have overlooked it, but it would be nice to explicitely state that the learned model is dependent for a specific robot and
has to be relearned for different robot kinematics.

**Quality Of The Limitations Section:**

Additional details required

**Reviewer Expertise:**

4: The reviewer is confident but not absolutely certain that the evaluation is correct

**Robotics Focus:**

Sufficient demonstration on hardware

**Strengths And Weaknesses:**

The paper describes the approach quite nicely and is easy to follow. Approach explanation and evaluation is
nicely done. The performance of the approach is very promising and seems to generalize quite well.

There is no mentioning of any time requirements for training, learning, or optimization at test time. This information
would be of great interest and a crucial detail to judge the performance and usability of the approach.

The limitations section mentions a few possible reasons for failure cases. A more differentiated discussion would have been
helpful. How often did a failure get caused by what problem? The failure rate is quite high even in simulation that
an actual discussion would be useful. Also a discussion on how/what could remedy these issues is missed and what
side-effects that would cause (e.g., increase input complexity at cost of more learning data needed).

**Summary Of Recommendation:**

The paper is easy to read, lays out the proposed approach clearly, and convincingly shows the performance with several experiments.
A few details were missing, but it is minor. I found the paper very informative and would like to see a continuation of this research line.
The approach also promises to be very useful for real robot manipulation.

---

> ### Author Response · Authors · 2022-08-26
> **Response to the Reviewer jW3a**
>
> We would like to thank the reviewer for taking the time to review our submission and for all the helpful suggestions for improving the manuscript.
>
> **Time and compute requirements**
>
> Thank you for pointing this out, we have added inference time details to the manuscript (lines 173-174) and training details to the supplementary material (lines 33-35 and 78-79). For our experiments, we used AMD Ryzen 9 5900X CPU and NVIDIA GeForce RTX 3080ti GPU. Training our Behaviour Cloning skills and affordance models took approximately 1 and 1.5 hours, respectively. During test time, solving the constrained optimisation problems to find starting robot configurations took on average 3.2 seconds. Note that during test time, **the optimisation problem needs to be solved only once using the initial scene observation.**
>
> **Limitations**
>
> We have extended the limitations section to better explain the scope and limitations of our approach (lines 284-289). In addition, to previously discussed limitations, we added the discussion about our approach's suitability for different types of skills. Our approach is most suitable for short-horizon manipulation skills that do not require a significant reorientation of the whole robot. However, the majority of manipulation skills are indeed short-horizon focusing on a particular task thus not significantly limiting the applicability of our approach. In addition, our method could be extended to work with skills that can avoid collision locally and still complete the task if suitably positioned at the start.
>
> **Dependency on specific robot kinematics**
>
> That is an excellent point, and we have added this to the revised manuscript (lines 101-104). In short, **the affordance model is dependent on the specific robot and its kinematics** as it needs to reason about the robot's geometry and its movement during skill execution. The a priori acquired skills, on the other hand, are not and they could be efficiently transferred to a different robot using the affordance model.

---

### Official Review · Reviewer_dTVn · 2022-08-01

**Originality:** Very Good
**Technical Quality:** Good
**Clarity Of Presentation:** Good
**Impact:** 4

**Recommendation:**

Weak Accept: I recommend accepting the paper, but will not argue for my recommendation if the majority of other reviewers have a different opinion.

**Summary:**

The paper proposed a novel heterogeneous graph-based affordance model to find the optimal robot configuration for executing the a priori acquired skill would complete the task, whilst avoiding collisions with the environment. The main contribution of the paper is an three-step approach that allows the trained policies trained in obstacle-free environments to be excuted in new cluttered environments wihtout any
collisions. They demonstrate that the proposed method can achieve good performance of the tasks on different tasks in challenging, cluttered and constrained environments.

**Issues:**

The result in the video shows that the method can learn to solve a particular task with two steps. Are those environement always  meet the assumption that  there exists a distribution of robot configurations from which an a priori learnt skill would be successfully executed?

It is confusing that, if a priori acquired skills are trained in a enviroment without constraints. Is there any cases that, after finding the starting configuration, the robot still can not successfully complete the task? As the a priori acquired skills can not avoiding the collision.

Also, how would the method work if a priori learnt skill would not be successfully executed?

Add discussion regarding if the reason that the proposed method outperform the end-to-end algorithm is because the a prior skill reduce the potential search area?


**Quality Of The Limitations Section:**

Limitations are addressed clearly

**Reviewer Expertise:**

3: The reviewer is fairly confident that the evaluation is correct

**Robotics Focus:**

Sufficient demonstration on hardware

**Strengths And Weaknesses:**

The idea of 3 steps task deployments that covers finding the starting configuration, move to the start configuration, execute the skill is interesting.

The idea of learning starting configuration with GNN is interesting.

The experiment result shows interesting behaviors learned by the proposed method. The comparison to prior methods in general supports the idea in this work.

It’s not clear if the proposed method can handle tasks with the environment that, the a prior learnt skill would not be successfully executed.

**Summary Of Recommendation:**

The paper presents a concrete learning algorithm and demonstrates good performance on a suite of difficult problems. However, the exposition of the paper could be improved and more discussions/comparisons would make the paper even stronger.

---

> ### Author Response · Authors · 2022-08-26
> **Response to the Reviewer dTVn**
>
> We would like to thank the reviewer for taking the time to review our submission and for all the helpful suggestions for improving the manuscript.
>
> **Assumptions and limitations**
>
> In this work, we assume that even in cluttered environments, there exists at least one robot configuration from which an a priori learnt skill would be successfully executed and would not collide with the surrounding obstacles. This assumption holds for the majority of environments and robotic manipulation skills. Manipulation skills acquired using methods such as Imitation Learning or Reinforcement learning are usually short-horizon and focus on a particular task, and, if started from a suitable configuration close to the target object, will not require significant movement of the whole robot. Thus, even if our method could not deal with cases when executing the skill without contact with the environment is impossible, it is still beneficial for the majority of environments and manipulation skills, not limiting the applicability of our approach. We have extended the limitations section to better explain the scope and limitations of our approach (lines 283-287).
>
> **Feasibility to execute skills in the evaluation environments**
>
> In our experiments, **we validate all the evaluation environments** ensuring that the robot can complete the task without colliding with the environment. We do so using an oracle used to collect demonstrations for our Behaviour Cloning (BC) skills. It doesn't take collision avoidance into account and, in the ideal case, if it can complete the task, so will BC skills. If an oracle can complete the task from at least one starting configuration, we use that environment in our evaluation.
>
> **Reasons for outperforming end-to-end policies**
>
> Indeed, end-to-end policies need to generalise across the whole workspace and learn how to complete the task, navigate free space and avoid obstacles. By using a priori acquired **skills that focus on completing the task** and a **motion planner to navigate the free space** and reach the predicted starting configuration, the search space is drastically reduced which results in a better performance. We have updated the manuscript (supplementary material) based on your suggestions and included a discussion regarding reducing the potential search space (lines 283-287).

---

### Official Review · Reviewer_rXLB · 2022-08-08

**Originality:** Good
**Technical Quality:** Good
**Clarity Of Presentation:** Very Good
**Impact:** 4

**Recommendation:**

Weak Accept: I recommend accepting the paper, but will not argue for my recommendation if the majority of other reviewers have a different opinion.

**Summary:**

This paper submission proposes a method to adapt *a priori* learned skills in open, obstacle-free environments to newly situated environments that contain clutter.  It is motivated by the fact that robots exist in complex and often cluttered environments, and the key hypothesis is that *adapting* a previously acquired skill to newly encountered clutter can prove more sample-efficient than learning from scratch in multiple, diverse cluttered environments.  The proposed method uses a graph-based affordance model with heterogeneous nodes, towards exploiting the underlying structure of the manipulation problem.

**Issues:**

See limitation listed above.

**Quality Of The Limitations Section:**

Additional details required

**Reviewer Expertise:**

3: The reviewer is fairly confident that the evaluation is correct

**Robotics Focus:**

Sufficient demonstration on hardware

**Strengths And Weaknesses:**

This paper addresses an important and relevant open problem in Robotics for the types of complex environments in which robots exist:  adapting object-centric control skills to execution in cluttered environments.  The method proposed is based upon an interesting and intuitive idea, leveraging a graph-based representation to exploit the inherent spatial relationships that exist between the robot, the target object (to be manipulated), and nearby obstacles present.  My primary questions are with regard to the evaluation.

Overall strengths and suggestions for improvement are summarised below.

More detailed strengths of paper:
+  The paper uses a principled and concise formalisation of the manipulation problem.  The representation captures important expressivity by including heterogeneous node and edge types and expected spatial structure that can be exploited by using an underlying graph representation.
+  Results in Table 2 show notable outperformance of the other tested methods in Environments 2 and up (where greater environment IDs map to increasingly complex environments).  The ablation experiments (section 4.4) add value, in characterising to some extent how the input representation impacts method efficacy.
+  There is some evaluation conducted on real robot hardware.
+  The paper is very well written.  The posing of the research questions to be investigated for each experiment was particularly useful in framing why that experiment was selected and how to interpret the results.

I would suggest the following to improve the paper submission:
-  I have questions around whether the set of baselines evaluated is sufficient. While some standard imitation learning methods (e.g Behavioural Cloning) are used for comparison, I would like to understand (and thus recommend explanation/discussion around) if there are more relevant SOA imitation learning or reinforcement learning baselines that actually aim to do constrained optimization or manipulation in cluttered/constrained environments?  If there are, why were those not tested as part of the Evaluation?
-  In Section 4.5 (Table 4), why not use all the same baselines as in Table 2?  For that matter, the same question applies for the results in Table 1.  In Table 4, what actually is the naive baseline and why is it the only one included?  Looking at Table 2, the naive baseline often does not outperform the other baselines.  BC seems to generally perform better; either way, results for BC are reported in the other experiments (Tables 1 and 2), so I would expect it to be included in the real world evaluation.  These questions around the baselines are important because this speaks to how useful the results and insights from the Evaluation are towards assessing the efficacy of the method and the value it adds to the set of existing/SOA approaches.

Additional minor comments:
In section 3.2.2 (page 5, line 139 and Equation 1), what is q?  Perhaps it is the robot starting configuration (as implied in line 138); however, I would recommend explicitly defining q, O, and y.
In Tables 1 and 2, would suggest **bolding** the best performing method(s) for each column.  This makes it easier to understand how the proposed method ranks comparably across tasks and environments.

**Summary Of Recommendation:**

I would recommend accepting this paper, based upon its contribution of a method for adapting acquired skills to new cluttered (and thus more complex environments).  To my knowledge, this is an important and still open problem in Robot Manipulation.

---

> ### Author Response · Authors · 2022-08-26
> **Response to the Reviewer rXLB**
>
> We would like to thank the reviewer for taking the time to review our submission and for all the helpful suggestions for improving the manuscript.
>
> **Choice of the baselines**
>
> Our objective was to investigate how skills which are acquired in unconstrained environments, can be transferred to new environments which are constrained. Robot manipulation skills are typically trained in unconstrained environments, because this is easier than in constrained environments. Usually, the focus is on learning a single task well, or learning a task which generalises across objects, rather than generalising across environments. Therefore, we are not trying to introduce a new method to learn a skill, because there is a range of popular methods for this; instead, we are introducing a method which takes existing skills, and enables them to work in new environments without altering them. **We are not aware of any published methods which also attempt to do this, which we could compare to**. If the reviewer is aware of any then we would be very interested to hear about these. We used Behaviour Cloning (BC) to acquire grasping and placing skills for our experiments, and so we compared our method to single end-to-end BC policies trained directly in cluttered environments, as this would be the natural extension to these policies if we wanted them to generalise across environments. We have extended the discussion and justification regarding the chosen baselines in the revised manuscript (supplementary material) (lines 102-104 and 122-125).
>
> **Real-world experiments**
>
> Our real-world experiments were designed to investigate whether our approach can be directly deployed on real hardware and perform well in challenging environments. Therefore, we tested our method in three different types of environments ranging from zero clutter to significant clutter. Running real-world experiments is very time-consuming. Therefore, we did not include all the baselines in this set of experiments. We chose to include the naive baseline as it is designed to **showcase that these environments are indeed extremely challenging**, and there is a need for learning starting configurations.
>
> **Used notation**
>
> q is, indeed, the starting configuration, from which the skill was rolled out. We have updated the manuscript based on your suggestions and explicitly defined the notation used in section 3.2.2 (lines 143-144).

---

### Author Response · Authors · 2022-08-26
**General Comment**

We would like to thank all reviewers for taking the time to review our submission and for all the insightful comments on how to improve it. We have taken your feedback and updated the manuscript and the supplementary material and uploaded the new versions (**changes are highlighted in green**).

**Additional details**

Based on the received suggestions we have added details on time and compute requirements, clarified used notation and included additional discussion about the performance of the method and justification of the chosen baselines.

**Baselines**

We have included further discussion and justification of the baselines used in our experiments in the supplementary material. We chose the baselines for our evaluation that allowed us for the most direct and fair comparisons. As we trained our grasping and placing skills using Behaviour Cloning (BC), we also used the same approach when comparing with end-to-end policies trained directly in constrained environments. All the other methods used the same BC policies without altering them during the execution. Due to this, we did not include any baselines that alter or re-learn said skill in constrained environments for a fair comparison.

**Limitations**

We have clarified the assumptions and limitations of our approach. Regardless of the discussed concerns, we believe that our method is still widely applicable, and all the mentioned issues could be mitigated in future work.

---

### Author Response · Authors · 2022-08-26
**Revised Supplementary Material**

Here we upload the revised version of the supplementary material.

---

### Meta-Review · Area_Chair_zr7X · 2022-08-15

**Recommendation:** Accept (Poster)
**Confidence:** 4

**Metareview:**

The authors propose a graph neural network model, which allows for training in obstacle-free environments and deploying collision-free policies in cluttered environments for manipulation tasks.  The authors show in both simulations and in the real world how this model can help improve the success rate of skill transfer. Their algorithm demonstrates good performance on a difficult problem.

The paper is original and could potentially represent a major advance in the field.  It is technically competent, with some additional analysis and missing details that need to be addressed.  It is generally clear and concise.   It has the potential to be impactful in the field and demonstrates the algorithm performing well in both simulation and reality.

However, the introduction needs rewriting to more strongly motivate the work.  Comments on scalability and generalizability need to be addressed – there is no discussion on time and compute requirements.  Limitations need to be more fully fleshed out, with more details required.  The experimentation needs expanding to allow for statistical significance testing to take place. Stronger motivation and explanation for the use of RL in this problem is required, as it seems that some of the ‘learned’ parts could be directly computed.  Some other experimental settings require explanation.  This should be accompanied with an expanded analysis that covers the above issues, as well as provides concrete reasoning for the observed performance differences.

***Update***

The authors were responsive to comments and have provided a strong paper worthy of acceptance.  The methodology itself still has question marks around its value w.r.t other approaches.

---

> ### Author Response · Authors · 2022-08-26
> **Response to the Area Chair zr7X**
>
> We would like to thank the Area Chair for facilitating the review process of our submission, a clear summary of reviews and insightful comments on improving the manuscript. We have also responded to each reviewer individually and addressed their concerns separately.
>
> **Motivation**
>
> Our main objective was to investigate how skills acquired in unconstrained environments can be transferred to new and cluttered ones. These skills can be acquired in any way and, in general, cannot be modelled. Changing or re-learning them in new environments can be extremely inefficient, therefore we opted for finding suitable starting configurations instead. We have updated the introduction to better get our motivation across.
>
> **Statistical significance**
>
> We have increased the number of replications of our simulation experiments using different random seeds to 6, doubling the number in the original manuscript to allow for statistical significance.
>
> **Scalability and generalizability**
>
> Our method was tested on a large number of randomised environments. Through 2 sets of experiments, we evaluated our method in unseen variations of scenes with a common underlying structure and significantly different structures from the ones used for training. We added additional examples of evaluation environments in the revised manuscript (supplementary material). Additionally, using skills acquired in unconstrained environments allowed us to retain their generalisability to novel objects without the need to re-learn them in constrained environments. This makes this overall approach significantly more scalable than methods which require explicitly training skills across different environments.
>
> **Time and compute requirements**
>
> We have updated the manuscript with additional details on time and compute requirements.
>
> **Limitations**
>
> We have extended the limitations section to better explain the scope and limitations of our approach. In addition to previously discussed issues, we have added a discussion about our approach's suitability for different types of skills. Discussed limitations do not significantly limit the applicability of our approach and could be mitigated in future work.
>
> **Experiment justification and discussion**
>
> We have updated the manuscript (supplementary material) to include a discussion about the performance compared to end-to-end policies and further justification of the chosen baselines.